# A Low FODMAP Diet Is Nutritionally Adequate and Therapeutically Efficacious in Community Dwelling Older Adults with Chronic Diarrhoea

**DOI:** 10.3390/nu12103002

**Published:** 2020-09-30

**Authors:** Leigh O’Brien, Paula Skidmore, Catherine Wall, Tim Wilkinson, Jane Muir, Chris Frampton, Richard Gearry

**Affiliations:** 1Department of Medicine, University of Otago, Christchurch 8140, New Zealand; paula.skidmore@otago.ac.nz (P.S.); catherine.wall@otago.ac.nz (C.W.); tim.wilkinson@otago.ac.nz (T.W.); chris.frampton@otago.ac.nz (C.F.); richard.gearry@otago.ac.nz (R.G.); 2Department of Gastroenterology, Central Clinical School, Monash University, Melbourne 3004, VIC, Australia; Jane.Muir@monash.edu

**Keywords:** fermentable oligosaccharides, disaccharides, monosaccharides, and polyols (FODMAP) diet, older adults, nutrition, quality of life, gastrointestinal diseases, diet, food, therapy

## Abstract

The low fermentable oligosaccharides, disaccharides, monosaccharides, and polyols (FODMAP)diet has been extensively researched, but not in the management of older adults with functional gastrointestinal symptoms. This study determines the positive and negative impacts of this dietary treatment in older adults with chronic diarrhea. A non-blinded intervention study was conducted with adults over 65 years with chronic diarrhea referred for colonoscopy where no cause was found. Participants followed a dietitian-led low FODMAP diet for six weeks and completed a structured assessment of gastrointestinal symptoms, the Hospital Anxiety and Depression scale, and a four-day food diary before and after the intervention. Twenty participants, mean age 76 years, were recruited. Adherence to the low FODMAP diet was acceptable; mean daily FODMAP intake reduced from 20.82 g to 3.75 g (*p* < 0.001) during the intervention and no clinically significant changes in macro- or micronutrient intakes were observed. There were clinically significant improvements in total gastrointestinal symptoms (pre diet 21.15/88 (standard deviation SD = 10.99), post diet 9.8/88 (SD = 9.58), *p* < 0.001) including diarrhea (pre diet 9.85 (SD = 3.84), post diet 4.05 (SD = 3.86), *p* < 0.001) and significant reductions in anxiety (pre diet 6.11/21 (SD = 4.31), post diet 4.26/21 (SD = 3.38), *p* < 0.05). In older adults the low FODMAP diet is clinically effective and does not jeopardise nutritional intake when supervised by an experienced dietitian.

## 1. Introduction

Hippocrates famously stated in the 5th century BC, “Let food be your medicine”. However, for many what they eat can result in unwanted gastrointestinal effects such as chronic diarrhea. Chronic diarrhea is common and associated with reduced quality of life [1,2], anxiety, and depression [3]. When functional it can be categorized as either diarrhea-predominant irritable bowel syndrome (IBS-D) when there is abdominal pain or functional diarrhea (FD) in the absence of abdominal pain [4,5]. Changes in diet, such as the low fermentable oligosaccharides, disaccharides, monosaccharides, and polyols (FODMAP) diet (LFD) [6,7,8,9], can resolve or reduce diarrhea. However, if not done under expert care, patients may not adequately follow the diet, learn how to identify trigger foods, or know how to ensure they maintain a nutritionally adequate balanced diet [10]. Interventional studies of the LFD have been safe and mostly positive, although children and older adults are yet to be studied [11]. In younger Western populations it is estimated that 10–20% of the population have IBS-D [12]. Recently, the Rome Foundation Global Study Group conducted a survey to estimate the prevalence of functional gastrointestinal disorders in over 70,000 adults from 33 different countries. They showed that 4.1% of older adults over the age of 65 have functional diarrhea [13]. The pathology of chronic diarrhea in older adults can be challenging, typically requiring consideration of colorectal cancer, polypharmacy [14], or age-related changes in gut physiology and microbiome.

Dietary manipulation and treatments in an older population carries risks. Older people have higher protein [15], calcium, and vitamin D requirements [16], are at risk of inadequate micronutrient intake [17,18], and 50% are at risk of malnutrition or are already malnourished [19]. Therefore, any dietary intervention that restricts the intake of specific foods may cause or exacerbate existing nutritional deficiencies. A French study found that older adults on a restrictive diet (low sodium, low cholesterol, or diabetic diet) were four times more likely to be at risk of malnourishment or be malnourished compared with age-matched controls [20]. The risk of becoming malnourished increased when participants followed more than one type of restrictive diet.

Given how safe and effective the LFD diet is in younger adults, it is essential to determine the positive and negative impacts of the LFD in an older population. This study aimed to determine whether an LFD is feasible, safe, effective, and acceptable for adults older than 65 years with chronic diarrhea.

## 2. Materials and Methods

### 2.1. Study Design and Patient Population

This was an unblinded, uncontrolled interventional observational study in adults older than 65 years. A formal power calculation could not be made as there are no published data concerning the efficacy or safety of an LFD in an older population. Ethical approval was obtained from the University of Otago Human Ethics (Health) Committee, Dunedin, New Zealand (H17/068) and all participants gave written informed consent.

General practice referrals to Christchurch Hospital Gastroenterology Department, Christchurch, New Zealand for investigation of unexplained diarrhea were screened. Inclusion criteria were: age > 65 years, diarrhea of greater than four weeks as the reason for referral, living in own home, they or a partner were able to prepare their own food, not already following an elimination diet, and no history of anxiety or stress that would exacerbate symptoms while following an LFD. Exclusion criteria were: type 1 or type 2 diabetes on medical treatment, known significant bowel disease (e.g., celiac disease, Crohn’s disease, ulcerative colitis, diverticular disease, colorectal cancer), previous bowel resection or, currently taking any medications likely to cause diarrhea (e.g., laxatives, Metformin). People were subsequently excluded if there was evidence of a non-functional cause of diarrhea found after colonoscopy or biopsies, such as bowel disease or microscopic colitis. Additionally, participants were asked to confirm with their general practitioner (GP) that coeliac disease has been eliminated as a potential cause of diarrhea before commencing the LFD.

All participants were reviewed by the research dietitian for an individualized initial appointment of 60–90 min, and a follow-up appointment of 30 min after six weeks of following the diet. At the initial appointment the mechanisms of the LFD were explained, food exclusions, foods allowed, and meal options were discussed. Participants were provided with written material explaining the LFD, written lists of high and low FODMAP foods, recipes, a suggested menu plan, a shopping guide, and a summary sheet of the main points discussed.

If symptoms had improved after six weeks of a strict LFD, participants were taught how to reintroduce each FODMAP carbohydrate group to determine the approximate portion size tolerated without triggering symptoms. A written reintroduction protocol was provided to each participant. If there was no improvement of symptoms, participants were advised to return to their previous diet.

### 2.2. Data Collection

On recruitment, demographic (age, sex, socioeconomic status (NZDep2013) [21]) and clinical data (medical history, weight, height, body mass index (BMI), prescribed medications) were collected. Prior to starting the LFD and six weeks later, gastrointestinal (GI) symptoms (assessed using the Structured Assessment of Gastrointestinal Symptoms Scale (SAGIS) [22]) and measures of anxiety and depression (assessed using the Hospital Anxiety and Depression Scale (HADS) [23]) were completed.

Participants completed a four-day food diary prior to the initial appointment and one week before the follow-up appointment. Written instructions were provided, and participants were asked to record all food and beverages consumed, how food was cooked, and the amounts eaten. A four-day food diary was chosen as it allows for a comparison of nutrient intake [24] before and during the LFD and to assess compliance. Participants’ perceptions of how the diet was taught, whether the resources provided were sufficient to follow the diet successfully, and any other feedback were recorded at the follow-up appointment.

### 2.3. Dietary Analysis and Data Entry

The food diaries were entered into and analyzed using FoodWorks 9 Professional Edition (version 9.0.3951, 2017, Xyris Software, Brisbane, Australia). New Zealand FOODfiles 2016 were used, which utilized Nutrient Reference Values for Australia and New Zealand. The main macronutrients (fat, carbohydrate and protein) along with total energy, total fiber, and micronutrients (calcium, folate, iron, iodine and magnesium) were calculated. FODMAP daily consumption was calculated using the Department of Gastroenterology, Monash University, Melbourne, Australia FODMAPs database and independently checked in case of any irregularities.

### 2.4. Statistical Analysis

Data are summarized as means and standard deviations (SD) or frequencies and percentages. Continuous data were compared between pre and post diet visits using paired *t*-tests. The rankings of specific health concerns were compared pre and post intervention using Wilcoxon signed rank tests. Dietary intake of macronutrients and micronutrients were compared against the Nutrient Reference Values for Australia and New Zealand [25]. Statistical significance was determined with a two-tailed *p*-value < 0.05.

All data collected were entered into IBM (SPSS v25, Armonk, NY, USA) for statistical analysis. The scores from the SAGIS data were entered in full and categorized into the five symptom clusters. The qualitative SAGIS questionnaire and end of intervention questionnaire data were coded thematically.

## 3. Results

### 3.1. Baseline Characteristics

A total of 442 referrals were screened between November 2017 and December 2018 of which 126 met the inclusion criteria, but 54 were excluded as colonoscopy or pathology results identified a non-functional cause for diarrhea. An invitation letter was sent to the remaining 72 patients with follow-up calls if needed. Twenty-three patients consented to take part in the study, with three subsequently withdrawing (Figure 1). The baseline characteristics of the 20 participants are shown in Table 1. The mean age was 76 years (range, 67 to 84 years). More females 15/20 (75%) than males took part and most (16/20) identified as New Zealand European. Ten (50%) participants were taking more than five regular medications per day and most 15/20 (75%) were from areas of low to medium deprivation (NZ deprivation index ≥ 5).

### 3.2. Digestive Symtpoms

At baseline the predominant symptoms were urgency to empty bowels, loose stools, excessive gas, pain prior to bowel motion, and diarrhea (Appendix A). After six weeks of the strict LFD (phase 1), there were significant improvements in the overall SAGIS score, as seen by a reduction in the score from pre to post diet. The pre diet mean score was 21.15/88 (SD = 10.99) compared to the post diet mean score of 9.8/88 (SD = 9.85) *p* < 0.001 (Table 2)** Significant improvements were observed in the domains of diarrhea/incontinence (pre diet 9.85 (SD = 3.84), post diet 4.05 (SD = 3.86) *p* < 0.001) where 18/20 (90%) of participants had an improvement, and epigastric pain/discomfort (pre diet 5.35 (SD = 4.67), post diet 2.45 (SD = 3.25) *p* < 0.001) where 12/20 (60%) experienced an improvement (Table 2, Figure 2 and Figure 3, Appendix A). For the diarrhea/incontinence domain 14/20 (70%) of participants experienced fewer loose bowel motions, 13/20 (65%) less diarrhea, and 13/20 (65%) less urgency to empty the bowel. In the acid/regurgitation domain, 11/20 (55%) of participants improved, mostly attributable to participants experiencing a decrease in belching/heart burn (Appendix A).

### 3.3. Health Concerns

Participants recorded their most and second most important health concern as part of the SAGIS questionnaire; 16 participants completed this question at baseline and follow-up. Thirteen participants reported that a GI symptoms such as diarrhea, GI pain, or variable bowel habit were an important concern for them at baseline. After completing phase 1 of the LFD, only seven participants reported a health concern relating to GI symptoms, the remaining participants reported that there were now no health concerns or replaced the health concern with a non-GI concern such as back pain. 

### 3.4. Depression and Anxiety

The HADS questionnaire was completed at baseline by 19/20 (95%) older adults; 17 had scores within the normal range, indicated by a score of less than seven, for depression and 12 were within the normal range for anxiety. There was a statistically significant reduction in both the scores for depression (pre diet 3.63/21 (SD = 2.67), post diet 2.11/21 (SD = 1.79), *p* < 0.01) and anxiety (pre diet 6.11/21 (SD = 4.31), post diet 4.26/21 (SD= 3.38), *p* < 0.05) between baseline and six-week follow-up. However, as most of the reductions were within the normal range these changes may not be clinically significant.

### 3.5. FODMAP Content of Dietary Intake

The mean FODMAP consumption reduced significantly during the dietary intervention (Table 3). At baseline lactose accounted for 69% of the total FODMAP intake. With lactose intake excluded, there was still a significant reduction in the consumption of the remaining FODMAP carbohydrates (pre diet 5.17 (SD = 3.51), post diet 1.76 (SD = 0.98), *p* < 0.001).

### 3.6. Nutrient Intake While Following a Strict Low FODMAP Diet

Macro- and micronutrient intake did not change significantly between baseline and follow-up (Table 4). At baseline, mean intake of protein and iron was greater than the recommended daily intake (RDI). However, for the remaining macro- and micronutrients most participants consumed less than the RDI and no participants met the RDI for calcium. On the LFD a third of participants consumed less than 20% of their baseline kilojoules (KJ), a third also consumed 20% less protein. Only two participants ate 10% less fiber from the baseline diet (Appendix A).

### 3.7. Qualitative Data

Nineteen participants provided feedback on the acceptability of the LFD. All participants agreed the diet was taught in a way that they could easily understand and 9/16 (56%) thought that it would be difficult to follow with only written information and no accompanying oral explanation or opportunity to ask questions (Appendix A). Participants were asked what they thought was the most difficult part of following the diet; only 5/19 (26%) participants reported that they missed restricted foods while 6/13 (46%) found the diet more expensive to follow (Appendix A).

## 4. Discussion

The LFD is a proven dietary approach to manage symptoms of IBS [6,7,26,27,28,29]. To the best of our knowledge, this is the first study to investigate the use of the LFD to treat chronic diarrhea in older adults. In this cohort of older adults, 90% experienced improvements in the SAGIS domain for diarrhea/incontinence, with 70% experiencing fewer loose bowel motions and 65% less diarrhea.

This compares favorably with studies of the LFD in younger adults where improvements are consistently seen in 50–80% of participants [8,29,30,31].

The LFD is defined as a dietary intake of less than 3.1 g of FODMAPs per day [30]. In this study, there was an 82% reduction in FODMAP intake indicating excellent dietary adherence with a mean reduction from 19.46 g to 3.75 g per day (1.76 g per day excluding lactose).

Dietary strategies to manage IBS symptoms, both non-evidenced (e.g., the paleo diet) and evidence-based approaches (LFD [32] and specific carbohydrate diet [9]) often focus on reducing bread and milk, which provide important nutrients for older adults [33] potentially leading to poor diet quality. Malnutrition is associated with poor outcomes for older adults [19,34]. Therefore, careful consideration of dietary adequacy is required prior to recommending dietary changes so that the risk of malnutrition does not outweigh the benefits of symptom relief. No participants experienced a significant change in nutrient intake during this study suggesting that, when carefully taught by an experienced dietitian, the LFD can be followed safely by older adults. There was, however, a non-significant reduction in calcium intake, with 40% of participants consuming less calcium. Many of these individuals already consumed less than the RDI. Suitable lactose free dairy alternatives were discussed, and all participants were encouraged to include at least two serves of calcium containing foods per day.

Other key nutrients for older adults are protein and fiber, there was no significant reduction in the mean intake across the group. There is inconsistency in the literature regarding changes in fiber on a low FODMAP diet [6,9,31,35]. Therefore, it is reassuring that in this study older adults are able to maintain their nutrients and fiber intake while following an LFD.

The baseline nutritional intake for some participants in this study was below the RDI for calcium, folate, and iodine, however intake did not significantly reduce further on the LFD. Poor nutritional intake has been observed in studies of people with IBS [36,37]. These observations suggest that those with gut symptoms possibly avoid specific foods, such as milk, fruit, and bread, which in turn could lead to lower intakes of calcium, folate, and iodine.

Furthermore, older adults often consume insufficient amounts of key nutrients such as calcium, folate, and magnesium [17,18,38]. Therefore, specific expert dietetic advice for older adults may be needed for key nutrient foods to ensure the safety of the LFD.

The LFD has been found to improve quality of life (QOL) in younger populations [39]. There are well-described associations between psychological and gastrointestinal symptoms in those with functional GI disorders through the gut–brain axis, with subsequent improvements in mental health with the resolution of GI symptoms [3,40,41] In this study, the HADS questionnaire and rating of health concerns (SAGIS questionnaire) were used to determine the impact of dietary changes on mental health and health concerns. There was no worsening of anxiety or depression while on the diet. Participants experienced changes in their health concerns with 67% no longer recording the same health concerns after the LFD intervention. Reporting of an IBS symptom as a primary health concern reduced from 81% to 43% of participants.

Older adults preferred verbal explanations to accompany written information about the LFD. In an randomized control trial (RCT), where the participants had a mean age of 38 years, Staudacher and colleagues found that symptom improvement for those in the LFD group was less than expected [8]. It was proposed that shorter face-to-face dietitian time and explanation of how the diet works may have resulted in reduced ‘buy in’ for those in the study. Verbal and individualized explanations from a dietitian experienced in teaching the LFD appears to be valued by participants regardless of age.

While the results of this study suggest that an LFD can improve functional symptoms in older adults with diarrhea and are consistent with studies in younger populations, there are limitations of the study. The sample was small (*n* = 20) and not formally powered to detect small changes. Study participants mostly identified as New Zealand European. Thus, the study findings may not be generalizable to all older New Zealanders. There may be expectation bias related to the non-blinded study design.

A four-day food record was used to gather nutrient and FODMAP data which is adequate to estimate macronutrients especially total energy and protein [24], fiber and some, but not all, micronutrient intakes. The questionnaire relies on self-reporting, a recognized limitation as food and quantities consumed may be under or over estimated. To improve the validity of the diet data, the same researcher checked diary items with the participants and entered all data into Foodworks 9 and the Monash FODMAP calculator. The questionnaires used in this study have been validated in younger populations, it is possible that the questionnaires may not have performed as well in this older population.

The study participants were asked to strictly follow phase one of the LFD. A modified approach that only excludes foods containing concentrated quantities of FODMAPs such as large portions of wheat, milk, yoghurt, legumes as well as onion, garlic, and some fruit is now recommended for those of high risk of malnutrition such as older adults [42]. This “FODMAP-gentle diet” approach works well under the supervision of an experienced dietitian, however, for the purpose of this pilot study, where the response of a group of unstudied individuals was taking place, it was considered that using the strict LFD approach would be a preferable method to determine whether older adults respond to the diet.

The major strength of this study is that it was conducted exclusively in older adults with diarrhea following a colonoscopy which excluded significant non-functional colonic pathology. To ensure that the LFD was implemented effectively, a dietitian designed and delivered comprehensive resources were used, including individualized low FODMAP meal and snack options. Although the study was small, all older adults completed the six-week dietary intervention, indicating that compliance for this age group is high and/or that participants found value in following the diet. The study assessed the impact of the strict LFD on GI symptoms, however, did not assess whether improvements were maintained during the reintroduction (phase 2) and personalization (phase 3) phases of the low FODMAP diet. Nor did it access the impact that the restriction of fermentable carbohydrates has on the microbiome, further research in this field is warranted.

As with other LFD studies, not all participants experienced improvements. This leaves the question of what else can be done for those who do not respond. Pathways for treatment of chronic functional diarrhea in adults, especially older adults, need to be developed further to include dietary strategies [14,43], and this study suggests that the LFD should be an important component to include in such pathways. Future research is needed to determine if older adults with chronic diarrhea are more likely to have poor nutrient intake (especially calcium, iodine, and folate) compared to those older adults without chronic diarrhea. Furthermore, clinicians need to be aware of pre-intervention dietary intake so that advice can be tailored to not only improve gut health but address any nutrient intake shortfalls.

## 5. Conclusions

In conclusion, this pilot study shows that the LFD, when dietitian led, is a feasible intervention in an older population, appears to be nutritional safe and improves GI symptoms.

## Figures and Tables

**Figure 1 nutrients-12-03002-f001:**
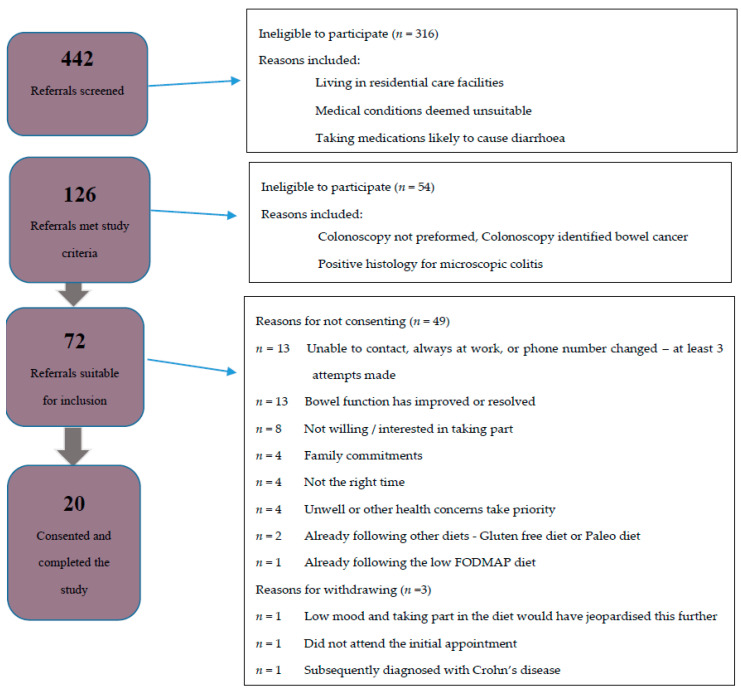
Flow diagram of participant recruitment. FODMAP: fermentable oligosaccharides, disaccharides, monosaccharides, and polyols.

**Figure 2 nutrients-12-03002-f002:**
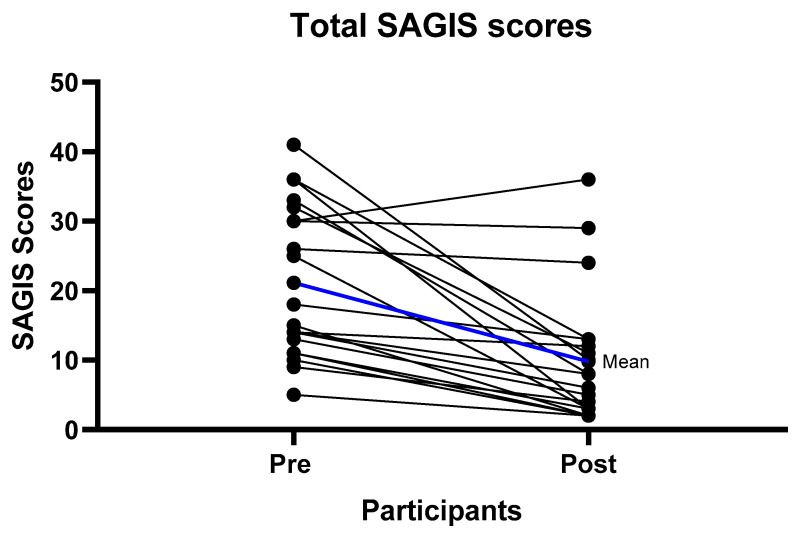
Individual gastrointestinal symptom improvement (Total Structured Assessment of Gastrointestinal Symptoms (SAGIS) scores) from baseline to follow-up. Mean difference from the pre intervention to follow-up scores was 11.35 with a standard deviation of 10.75 *p* < 0.001.

**Figure 3 nutrients-12-03002-f003:**
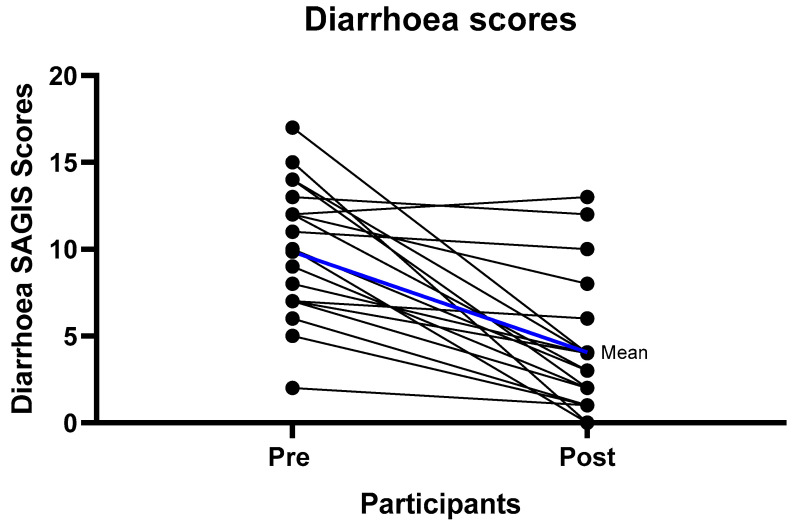
Individual diarrhea improvement scores from baseline to follow-up. Mean difference from the pre intervention to follow-up scores of 5.8 with a standard deviation of 4.48 *p* < 0.001.

**Table 1 nutrients-12-03002-t001:** Demographic characteristics of twenty study participants that were included in the pilot study.

Characteristic	Participants
Mean age (years)	76 (67–84)
Gender	
Female Male	15 (75%) 5 (25%)
Ethnicity	
New Zealand European European Other	16 (80%) 2 (10%) 2 (10%)
Deprivation index	
<5 ≥5	5 (25%) 15 (75%)
Number of medications	
<5 ≥5	10 (50%) 10 (50%)

**Table 2 nutrients-12-03002-t002:** Mean difference for Structured Assessment of Gastrointestinal Symptoms (SAGIS) domains from baseline to follow-up. Comparisons were made by two -tailed paired samples *t*-test. Data shown represents the mean difference and standard deviation (SD).

SAGIS Domains	Pre-Intervention Symptom Score (*n* = 20) Mean (SD)	Post-Intervention Symptom Score (*n* = 20) Mean (SD)	Mean (S) Difference	*p*-Value
Acid regurgitation/gas Three items Maximum score = 12	1.8 (1.20)	0.9 (1.83)	0.9 (1.71)	<0.05
Diarrhea/incontinence Six items Maximum score = 24	9.85 (3.84)	4.05 (3.86)	5.8 (4.48)	<0.001
Difficult defecation and constipation Two items Maximum score = 8	1.7 (1.78)	1.05 (1.39)	0.65 (1.84)	0.131
Nausea/vomiting Four items Maximum score = 16	0.75 (1.52)	0.35 (0.75)	0.4 (1.40)	0.214
Epigastric pain/discomfort Seven items Maximum score = 24	5.35 (4.67)	2.45 (3.25)	2.9 (3.99)	<0.005
TOTAL Maximum score = 84	21.15 (10.99)	9.8 (9.58)	11.36 (10.76)	<0.001

**Table 3 nutrients-12-03002-t003:** Mean daily fermentable oligosaccharides, disaccharides, monosaccharides, and polyols (FODMAP) intake for each FODMAP group and totals at baseline and follow-up appointment. Comparisons were made by two-tailed paired samples *t*-test. Data shown represents the percentage change, mean difference and SD.

Intake Per Day	Pre-Intervention Mean (SD)	Post-Intervention Mean (SD)	Percentage Change of Intake (%)	Mean Difference (SD)	*p*-Value
Excess fructose (g)	1.37 (1.57)	0.18 (0.29)	87%	1.19 (1.54)	<0.001
Oligosaccharides (g)	2.66 (1.01)	1.26 (0.69)	53%	1.39 (1.15)	<0.001
Sorbitol (g)	1.92 (2.93)	0.22 (0.28)	89%	1.70 (2.96)	<0.001
Mannitol (g)	0.59 (1.28)	0.09 (0.19)	85%	0.50 (1.30)	<0.001
Lactose (g)	14.29 (9.81)	1.99 (3.33)	86%	12.29 (9.51)	<0.001
Total FODMAPs (g)	19.46 (9.96)	3.75 (3.78)	82%	15.88 (9.68)	<0.001
Total FODMAPs without lactose (g)	5.17 (3.51)	1.76 (0.98)	73%	1.58 (0.90)	<0.001

Fructose is absorbed in the presence of glucose; when there is more fructose than glucose present in a food it is referred to as excess fructose, it is the excess fructose that, for some, is not absorbed in the small intestine resulting in gastrointestinal (GI) symptoms.

**Table 4 nutrients-12-03002-t004:** Mean nutrient intake for participants at baseline and follow-up appointment. Comparisons were made by two-tailed paired samples *t*-test. Data shown represents the mean difference and SD.

Nutrient	RDI	Pre-Intervention Mean (SD)	Post-Intervention Mean (SD)	Mean Difference (SD)	*p*-Value
Total Energy (KJ/kcal) Male Female	IR IR	8237/1969 (2552/610) 6980/1668 (1367/327)	8474/2025 (3569/853) 6685/1597 (1455/348)	238/57 (1424/340) 295/71 (1146/274)	0.728 0.335
Protein (g) Male Female	81 57	83.6 (16.7) 72.8 (14.5)	87.7 (28.6) 71.7 (13.4)	4.1 (16.3) 1.1 (14.3)	0.604 0.766
Fiber (g) Male Female	30 25	26.6 (7.7) 22.9 (7.0)	25.5 (3.4) 22.2 (6.3)	1.1 (7.1) 0.7 (6.9)	0.749 0.714
Calcium (mg) Male Female	1300 1300	947.2 (363.4) 935.3 (204.2)	967.6 (514.2) 890.3 (280.7)	20.4 (311.9) 44.9 (234.1)	0.891 0.470
Folate (µg) Male Female	400 400	259.6 (108.1) 300.3 (91.9)	333.0 (178.2) 318.2 (102.8)	73.4 (82.4) 18.0 (137.5)	0.059 0.620
Iron (mg) Male Female	8 8	12.2 (1.5) 10.8 (2.9)	13.4 (4.4) 11.4 (4.7)	1.2 (4.1) 0.6 (4.6)	0.547 0.643
Iodine (µg) Male Female	150 150	60.0 (18.5) 55.7 (18.0)	89.4 (39.8) 62.7 (48.0)	29.4 (30.3) 7.0 (53.2)	0.096 0.618
Magnesium (mg) Male Female	420 320	345.0 (91.2) 306.4 (87.3)	371.6 (146.0) 315.0 (91.6)	26.6 (62.9) 8.6 (82.4)	0.398 0.692

RDI = Recommended daily intake, IR = Individual requirement, KJ = kilojoule, g = gram, µg = micrograms, mg = milligrams.

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
