# Peer review of "A Low FODMAP Diet Is Nutritionally Adequate and Therapeutically Efficacious in Community Dwelling Older Adults with Chronic Diarrhoea"

_nutrients, 2020, doi:10.3390/nu12103002_

Round 1

Reviewer 1 Report

This is a nice pilot study that helps fill a gap in the literature about the role and clinical benefit and nutritional adequacy of a low FODMAP diet (LFD) in the elderly patient with diarrhoea. I think it has been well presented and the discussion was good. Overall, a nice contribution to the field.

As acknowledged by the authors this is a preliminary single-centre study involving a small number of participants with an open-label design and clearly further validation studies in larger numbers and populations will be needed. I have no problems with this as I think the findings still provide a very helpful basis to inform future studies.

My queries are minor and mostly relate to clarifying the patient population studied and how they have been defined in this paper.

  1. This study included the independent elderly (> 65) patient with diarrhoea (> 4 wks) who was referred to the Gastroenterology Department.
  • I am a little unsure how patients were worked-up. It states ALL patients had colonoscopy and 54 were excluded due to colonoscopy or other pathology showing organic findings. What was this other work-up undertaken?
  • Coeliac disease was an exclusion criterion. Was it excluded based on formal coeliac serology or a review of medical records? Undiagnosed coeliac disease in the elderly may be a cause of chronic diarrhoea and may positively respond to a LFD due to the concomitant reduction in wheat.
  • I note that the Study Flow (Figure 1) showed 54 people were ineligible to participate because “colonoscopy was not performed”. However, this is not consistent with the manuscript text which states it was because an organic cause was found on colonoscopy or pathology results. Please clarify in the manuscript.
  • What time frame was the recruitment period over?
  1. The assumption is that the responders have a functional cause for their diarrhoea. Therefore, it would be of interest to report how many of the participants met formal ROME criteria for IBS.
  2. “The study assessed the impact of the strict LFD on GI symptoms, however did not assess whether improvements were maintained during the reintroduction (phase 2) and personalisation (phase 3) phases of the low FODMAP diet.”. Could also mention that in additional to lack on data on long-term effects, impact on microbiome (that could be potentially adversely affected by LFD) was not assessed. (Totally understandable for this study, so not a criticism).

Author Response

22nd September 2020

Leigh O’Brien
Department of Medicine

University of Otago – Christchurch

PO Box 4345

Christchurch 8140

New Zealand

Dear reviewers

Thank you for taking the time to review the manuscript entitled “A low FODMAP diet is nutritionally adequate and therapeutically efficacious in community dwelling older adults with chronic diarrhoea” for the special edition “Nutritional Pearls and Pitfalls of Gastrointestinal Diseases”.  

Your review has been most helpful in revising the manuscript. We have made the following changes as outlined under each bolded comment.

I am a little unsure how patients were worked-up. It states ALL patients had colonoscopy and 54 were excluded due to colonoscopy or other pathology showing organic findings. What was this other work-up undertaken?

Biopsies were taken at the time of the colonoscopy, patients were excluded if the biopsy indicated microscopic colitis. We have now added that detail into the methods.

Coeliac disease was an exclusion criterion. Was it excluded based on formal coeliac serology or a review of medical records? Undiagnosed coeliac disease in the elderly may be a cause of chronic diarrhoea and may positively respond to a LFD due to the concomitant reduction in wheat.

Those with known coeliac disease were excluded during the screening process. Participants were then asked at the initial appointment if they has been screened for coeliac disease, and if not recommended that this was done before commencing the diet. We have now added this detail into the methods.

I note that the Study Flow (Figure 1) showed 54 people were ineligible to participate because “colonoscopy was not performed”. However, this is not consistent with the manuscript text which states it was because an organic cause was found on colonoscopy or pathology results. Please clarify in the manuscript.

Unfortunately the reasons for ineligible to participate were cut off from the figure during to formatting, this has been resolved.

What time frame was the recruitment period over?

The time for recruitment was from November 2017 to December 2018. This included the time for screening, colonoscopy to take place, results to be formalised and patient to be contacted by the research dietitian. The time frame is reported in the results section.

The assumption is that the responders have a functional cause for their diarrhoea. Therefore, it would be of interest to report how many of the participants met formal ROME criteria for IBS.

Yes this would have been interesting to see this. We decided to include all those with chronic diarrhoea, but as you mentioned a diagnosis of IBS could have been made for some of the participants.  

“The study assessed the impact of the strict LFD on GI symptoms, however did not assess whether improvements were maintained during the reintroduction (phase 2) and personalisation (phase 3) phases of the low FODMAP diet.”. Could also mention that in additional to lack on data on long-term effects, impact on microbiome (that could be potentially adversely affected by LFD) was not assessed. (Totally understandable for this study, so not a criticism).

The reintroduction of FODMAP sugar groups back into the diet (Phase 2) and long term management (Phase 3) are very important aspects of the low FODMAP diet, we would like to study this in future studies along with the gut microbiome.   We have added a comment into the discussion regarding the impact on the microbiome not being studied.

Once again thank you for your consideration of this manuscript.

Sincerely,

Leigh O’Brien

Reviewer 2 Report

  1. authors do not mention patients other comorbidities and medication but they mention that they record medications. what are the other comorbidities of patients and maybe those resulted in this nutritional status of patients also? (a little mention in discussion perhaps?) 
  2. why authors used just SAGIS and not ROME criteria also? They do not mention diagnosis for diarrhea, they excluded organic cause very well but it should be noted if diarrhea could be classified as IBS or functional based on ROME criteria. This would lead to be more homogenous towards other studies that examine gastrointestinal problems and FODMAPS and be more specific of the cause for diarrhea. 
  3. in materials and methods section in 2.3 section , lines 108-110 should be moved to section 2.4
  4. in results section 3.2 digestive symptoms , quantitive results should be a little better presented because including responders of patients and mean and standard deviation in one sentence is a bit confusing to reader. lines 131-135 need a better phrasing because is not quite understandable what they say.
  5. more elaboration and details on health concerns page 4 lines 141-143. what are health concerns , for what reason. 
  6. results section depression anxiety ,provide numbers for normal range of depression and anxiety in HADS
  7. line 156 should be italic like the others 
  8. energy intake to be mentioned in kcal also
  9. in flow diagram in the two squares something is missing because of formatting, please check
  10. tables should be a little better looking, try a little better formatting
  11. in table 1 what is a deprivation index? 1st mention in whole study. 
  12. in table 2 are there any cutoffs for the scores? maybe include a mention in results about the scale regarding for example the higher the better? or low is worst?
  13. what authors mean with excess fructose in table 3? 
  14. In table 4 p values although are not significant would look better if they were with numbers
  15. in discussion page 10 , lines 223-226 needs better phrasing to be more understandable
  16. lines 241-244 better phrasing in milk -calcium, fruit-folate, bread-iodine to be more concise.
  17. lines 256-258  mean age should be mentioned in better phrasing  because the sentence is cut in middle right now  
  18. lines 284-286 needs article in "dietitian designed..." 

Author Response

22nd September 2020

Leigh O’Brien
Department of Medicine

University of Otago – Christchurch

PO Box 4345

Christchurch 8140

New Zealand

Dear reviewer

Thank you for taking the time to review the manuscript entitled “A low FODMAP diet is nutritionally adequate and therapeutically efficacious in community dwelling older adults with chronic diarrhoea” for the special edition “Nutritional Pearls and Pitfalls of Gastrointestinal Diseases”.  

Your review has been most helpful in revising the manuscript. We have made the following changes as outlined under each bolded comment.

authors do not mention patients other comorbidities and medication but they mention that they record medications. what are the other comorbidities of patients and maybe those resulted in this nutritional status of patients also? (a little mention in discussion perhaps?)

As part of the exclusion process, we excluded those taking medications likely to cause diarrhoea and also those with medical conditions deemed unsuitable. This information was missing from figure 1 due to a formatting issue but is now resolved. We have included a comment on the number of medications taken in the results under 3.1.

Why authors used just SAGIS and not ROME criteria also? They do not mention diagnosis for diarrhea, they excluded organic cause very well but it should be noted if diarrhea could be classified as IBS or functional based on ROME criteria. This would lead to be more homogenous towards other studies that examine gastrointestinal problems and FODMAPS and be more specific of the cause for diarrhea.

We were interested in those with chronic diarrhoea and not just IBS or functional diarrhoea. The ROME criteria includes information on pain, also included in the SAGIS and consistency of bowel motions, likewise also included in the SAGIS questionnaire. The additional criteria in the ROME criteria is based on a duration of more than three months, as the time from initial referral for colonoscopy to date of inclusion into the study is longer than three months we did not need to include this criteria. We believe that the SAGIS questionnaire is adequate for the extent of bowel symptoms we investigated and degree of change from being on the diet.  

in materials and methods section in 2.3 section , lines 108-110 should be moved to section 2.4

Thank you. These lines have now been moved to section 2.4

in results section 3.2 digestive symptoms , quantitive results should be a little better presented because including responders of patients and mean and standard deviation in one sentence is a bit confusing to reader. lines 131-135 need a better phrasing because is not quite understandable what they say.

Thank you we have changed it to read The pre-diet mean score was 21.15/88 (SD=10.99) compared to the post diet mean score of 9.8/88 (SD=9.85) p < 0.001) (Table 2)

more elaboration and details on health concerns page 4 lines 141-143. what are health concerns , for what reason.

This has been rewritten to give more clarity

results section depression anxiety ,provide numbers for normal range of depression and anxiety in HADS

The HADS questionnaire was completed at baseline by 19/20 (95%) older adults; 17 had scores within the normal range, for depression (less than 7) and 12 were within the normal range for anxiety (less than xxxx?). There was a statistically significant reduction in both the scores for depression (pre diet 3.63/21 (SD=2.67), post diet 2.11/21 (SD=1.79), p < 0.01) and anxiety (pre diet 6.11/21 (SD= 4.31), post diet 4.26/21 (SD= 3.38), p < 0.05) between baseline and six-week follow up. However, as most of the reductions were within the normal range these changes may not be clinically significant.

line 156 should be italic like the others

Thank you this has now been changed

energy intake to be mentioned in kcal also

Thank you this has now been changed

in flow diagram in the two squares something is missing because of formatting, please check

Thank you this has now been changed

tables should be a little better looking, try a little better formatting

Thank you the tables have been changed

in table 1 what is a deprivation index? 1st mention in whole study.

A sentence has been added into 3.1 Baseline characteristics.

in table 2 are there any cutoffs for the scores? –maybe include a mention in results about the scale regarding for example the higher the better? or low is worst?

There is no cut off as such, a reduction of scores indicates an improvement of symptoms. We have included this in the results - line 140

what authors mean with excess fructose in table 3?

Fructose is absorbed in the presence of glucose; when there is more fructose than glucose present in a food it is referred to as excess fructose, it is the excess fructose that, for some, is not absorbed in the small intestine resulting in GI symptoms. We have added this as footnote to Table 3

In table 4 p values although are not significant would look better if they were with numbers

The p-values have been added to the tables

in discussion page 10 , lines 223-226 needs better phrasing to be more understandable

As recommended, we have now change the passage to say: The LFD is a proven dietary approach to manage symptoms of IBS [6,7,26-29]. To the best of our knowledge, this is the first study to investigate the use of the LFD to treat chronic diarrhea in older adults. In this cohort of older adults, 90% experienced improvements in the SAGIS domain for diarrhea/incontinence, with 70% experiencing fewer loose bowel motions and 65% less diarrhea.

lines 241-244 better phrasing in milk -calcium, fruit-folate, bread-iodine to be more concise.

As recommended, we have now change the passage to say: These observations suggest that those with gut symptoms possibly avoid specific foods, such as milk, fruit and bread, which in turn could lead to lower intakes of calcium, folate and iodine.

lines 256-258 mean age should be mentioned in better phrasing because the sentence is cut in middle right now

This paragraph has now been changed to:

Older adults preferred verbal explanations to accompany written information about the LFD. In a RCT, where the participants had a mean age of 38 years, Staudacher and colleagues found that symptom improvement for those in the LFD group was less than expected [8]. It was proposed that shorter face-to-face dietitian time and explanation of how the diet works may have resulted in reduced ‘buy in’ for those in the study. Verbal and individualised explanation from a dietitian, experienced in teaching the LFD, appears to be valued by participants regardless of age.

lines 284-286 needs article in "dietitian designed..."

Thank you, this his has been corrected - line 325

Once again thank you for your consideration of this manuscript.

Sincerely,

Leigh O’Brien